# Fast Machine Learning with Byzantine Workers and Servers

## Abstract

Machine Learning (ML) solutions are nowadays distributed and are prone to various types of component failures, which can be encompassed in so-called Byzantine behavior. This paper introduces LiuBei, a Byzantine-resilient ML algorithm that does not trust any individual component in the network (neither workers nor servers), nor does it induce additional communication rounds (on average), compared to standard non-Byzantine resilient algorithms. LiuBei builds upon gradient aggregation rules (GARs) to tolerate a minority of Byzantine workers. Besides, LiuBei replicates the parameter server on multiple machines instead of trusting it. We introduce a novel filtering mechanism that enables workers to filter out replies from Byzantine server replicas without requiring communication with all servers. Such a filtering mechanism is based on network synchrony, Lipschitz continuity of the loss function, and the GAR used to aggregate workers' gradients. We also introduce a protocol, *scatter/gather*, to bound drifts between models on correct servers with a small number of communication messages. We theoretically prove that LiuBei achieves Byzantine resilience to both servers and workers and guarantees convergence. We build LiuBei using TensorFlow, and we show that LiuBei tolerates Byzantine behavior with an accuracy loss of around 5% and around 24% convergence overhead compared to vanilla TensorFlow. We moreover show that the throughput gain of LiuBei compared to another state–of–the–art Byzantine–resilient ML algorithm (that assumes network asynchrony) is 70%.

## 1 Introduction

Scaling Machine Learning (ML) algorithms with existing datasets and models (21) calls for distributed solutions (25; 27; 12). The standard distribution approach nowadays is the parameter server architecture (26). Such an architecture employs two types of machines: workers and parameter servers. Typically, workers perform the model update computation, following today's standard workhorse algorithm for ML: stochastic gradient descent (SGD). The parameter server updates the model in each iteration after aggregating gradients from workers. Based on this, the typical iteration execution includes two communication rounds: workers pull a model from the server, which in turn then pulls gradients from workers.

However, such a solution is prone to a various amount of failures of the distributed system components. Centralizing the parameter server in one machine makes it prone to a crash failure that can then stop the entire learning procedure. Various kinds of failures, including software bugs, hardware defects, or even hacked machines, can lead to corrupted gradients, diverging the learning process (6; 35). Such behavior can be abstracted in the most general form of failures, namely Byzantine failures (23), modeling the very fact that a machine in the system behaves arbitrarily. Given the increasing use of ML in mission-critical applications (17; 30), tolerating such failures is crucial.

Byzantine ML solutions so far mostly focused on Byzantine workers, ignoring the possibility of Byzantine servers. Typically, a statistically-robust gradient aggregation rule (GAR) is employed by the parameter server to aggregate gradients received from workers, rather than using the vulnerable averaging of gradients. Such a GAR e.g., Krum (8), Median (34) is statistically proven to guarantee convergence of the learning processes even in the presence of a minority of Byzantine workers, under some assumptions on the correct workers.

GuanYu (15) is, so far, the only proposal that solves the *total*[1] Byzantine ML problem. GuanYu uses a GAR, e.g., Bulyan (16) for aggregating workers' gradients and Median (34) for aggregating models received from servers. Since it uses a GAR (i.e., Median) for aggregating models, it requires workers to communicate with a majority of servers in each iteration. Moreover, GuanYu assumes *network asynchrony*, namely that there is no bound on communication delays. But this comes with a cost: First, it requires three communication rounds, instead of two in the vanilla case, and it requires an assumption on the maximum distance between parameter vectors, i.e., models at correct servers, which might be hard to obtain in certain cases. Also, GuanYu requires a large number of compute nodes and server replicas to work, as one cannot differentiate between a Byzantine machine and a slow one in such network (18).

Having a *synchronous network* is common in practice for many distributed environments (19; 36) as practitioners can usually predict a conservative upper bound on the network delays. Assuming network synchrony allows tolerating Byzantine failures using the standard solutions of state machine replication (SMR), e.g., (22). But SMR in the ML context raises a few concerns: First, SMR requires multiple communication rounds per iteration for agreement on the current state of the model and the updating step. Given the typical sizes of models nowadays, the overhead of communication rounds is big enough to hide the effectiveness of distributing the learning task (19; 20). Second, SMR requires replicated workers that use the same data batch (11). This reveals problematic in the case of using private data that cannot be shared with others (32).

In this paper, we present LIUBEI, a total Byzantine–tolerant ML algorithm that does not trust any individual component, while almost not adding any communication overhead, compared to standard non–Byzantine deployments. LIUBEI introduces a novel mechanism that filters out replies from Byzantine servers without requiring to communicate with all servers. Moreover, LIUBEI introduces a novel protocol to reduce the number of communication rounds. Essentially, LIUBEI operates in two phases: *scatter* and *gather*. In the *scatter* phase, servers work independently and do not communicate with each other. In the *gather* phase, correct servers communicate to bring their view of models back close to each other. The number of *gather* steps is usually very small and hence, their overhead is insignificant. We prove that LIUBEI tolerates Byzantine failures and guarantees learning convergence.

LIUBEI uses a GAR to aggregate workers' gradients and hence, tolerates Byzantine gradients. Tolerating Byzantine servers relies on our filtering technique and the *scatter/gather* protocol; both assume network synchrony. In each iteration in the *scatter* phase, each worker pulls the updated model from only one server and then uses two filters to check the legitimacy of such a model: the Lipschitz filter and the models filter. On the one hand, the Lipschitz filter checks the growth of the model with respect to its gradient. A Lipschitz value is computed for a received model, which is then compared to Lipschitz values of other correct models. On the other hand, the models filter bounds the difference between models in two consecutive rounds. Based on the guarantees given by the used GAR, a worker can speculatively compute an updated model that should be close to the received model. Bounding the difference between both models allows workers to suspect whether a received model is Byzantine or not. After $T$ *scatter* iterations, LIUBEI does one *gather* step with the goal of bringing the models on correct servers back close to each other. Hence, LIUBEI does not require an assumption on the maximum distance between two correct models, as in GuanYu (15).

We implement LIUBEI over TensorFlow to show its performance on typical workloads. We show that LIUBEI can tolerate both Byzantine servers and workers while guaranteeing convergence. We also show that LIUBEI can achieve the same performance as GuanYu, with 5% accuracy loss compared to vanilla TensorFlow. We quantify LIUBEI overhead, compared to vanilla TensorFlow, to around 24% and with a throughput gain of 70% compared to GuanYu. Our source code is available (4). The passphrase for decrypting the code is: <JFZ2a+}QcDfDw4uM]LSWSrtt$x;}7 Rj.y3KfmF .

## 2 MODEL AND ASSUMPTIONS

### 2.1 BYZANTINE MACHINE LEARNING

The Byzantine problem was first introduced in the context of distributed services (23) in which a minority of the distributed system components could behave arbitrarily. Such a behavior could be the result of software bugs, hardware defects, message omissions, or even hacked machines. Similarly, the Byzantine failure in a distributed ML system occurs when one of the components in such a

---

[1]We use the term *total Byzantine resilience* to denote resilience against both workers and parameter servers.

system behaves arbitrarily, i.e., not following the algorithm. For example, a Byzantine machine can send a biased estimate of a gradient to another machine, which leads to a corrupted learning model accordingly or even learning divergence (6). Byzantine failures also abstract the *data poisoning* problem (7), which happens when a machine owns maliciously labeled data (i.e., misleading data); this may result in learning a corrupted model especially–crafted by the adversary. If the learning is centralized in one machine (as with the standard parameter server architecture (26)), things can go worse if such a machine is Byzantine as in this case, the adversary (which hacked such a machine) can write whatever it wants to the final model, making the learning process almost useless. Such behavior should be tolerated as ML is nowadays used in mission–critical applications.

## 2.2 System Model

We consider the standard parameter server architecture (26), where a logically centralized *server* holds the learning parameters, and *workers* do the gradient computation task. In each learning step, workers pull the latest version of the learning model from the parameter server and then, towards the end of the iteration, the parameter server aggregates the gradients computed by workers and updates the model. We assume synchronous training: both the parameter server and workers synchronize their iteration number. We also assume a synchronous communication model and therefore, we assume an upper bound on communication and computation delays. Thus, all machines expect a response from

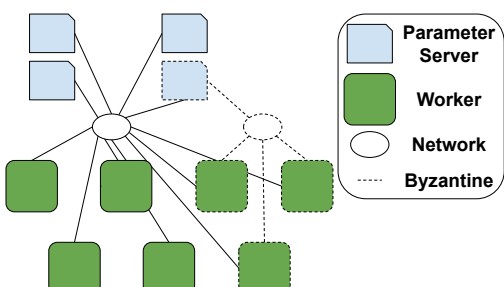

Figure 1: A setup with 7 workers (3 are Byzantine) and 4 servers (1 is Byzantine). Byzantine machines form together a single adversarial entity, where they communicate over their own covert network.

other machines within this bound, and non–responding machines can be safely flagged crashing.

Unlike most of the existing work in Byzantine ML literature, we do not assume a trusted parameter server. We replicate the parameter server on $n_{ps}$ machines, among which up to $f_{ps}$ machines could be Byzantine, i.e., behave arbitrarily, with $n_{ps} \geq 3f_{ps} + 1$. In addition, we assume that up to $f_w$ out of $n_w$ workers could be also Byzantine with $n_w \geq 2f_w + 1$. Figure 1 depicts such a setup.

The goal of the Byzantine nodes, be they workers or servers, is to impair the learning, by making it converge to a state different from the one that would have been obtained if no adversary had stymied the learning process. Byzantine nodes can cooperate to achieve this goal. We assume these nodes have unbounded computation power and arbitrarily fast communication channels (covert network in Figure 1). We also assume these nodes have access to the full training dataset and can overhear gradients sent by the correct workers. We assume that honest nodes can authenticate the source of a message so as to prevent spoofing and Sybil attacks (10).

We follow the usual assumptions in the Byzantine ML literature (11; 8; 16) (see supplementary material, Section 3.1). We assume that the training data is identically and independently distributed (*iid*) on workers (24) so that they can compute, at step $t \in \mathbb{N}$, an unbiased estimate ($g_t$) of the true gradient $\nabla L(\theta)$ (where $\theta$ represents the learning model of size $d$) with a sufficiently low variance.

## 3 LiuBei

In this section, we describe LiuBei algorithm. First, we describe how we tolerate Byzantine workers and servers, explaining our filtering technique and providing an intuition on why it works. Then, we present the training loop of both servers and workers.

### 3.1 Total Byzantine Resilience

**Byzantine workers.** Tolerating Byzantine workers is well-studied in the ML literature. Usually, the parameter server uses a statistically robust *gradient aggregation rule (GAR)*, e.g., (34; 16) that ensures having a correct gradient despite the presence of a minority of Byzantine gradients. LiuBei can use any existing *synchronous* GAR that follows the robustness definition of (8), which is also formally given in the supplementary material, Section 1.1. We choose *MDA* (31) for our algorithm as it gives practical resilience guarantees (supplementary material, Section 4) with a reasonable overhead as confirmed in our evaluation (Section 4). *MDA* requires $n_w \geq 2f_w + 1$.

**Byzantine servers.** Tolerating Byzantine servers using robust aggregation requires communication with all servers in each round, which results in a big communication overhead. We follow another route: We let each worker pull only one model from any of the server replicas and then checks if the pulled model is *suspicious* or not. A worker does this check by applying two filters on the pulled model (the *Lipschitz* filter and the *models* filter), which we describe below. If the model is suspicious, the worker discards it and pulls a new model from another parameter server. Thus, the maximum number of models that can be pulled by a worker in one iteration is $f_{ps} + 1$.

**Lipschitz filter.** Based on the standard Lipschitz continuity of the loss function assumption (9; 8), previous work uses empirical estimations for the Lipschitz coefficient to filter out gradients (from Byzantine workers) in asynchronous learning (14). We use a similar idea, yet to filter out *models* (on the workers' side) from Byzantine *servers*. The filter works as follows: Assume a worker $j$ that owns a model $\theta_t^{(j)}$ and a gradient it computed $g_t^{(j)}$ based on that model at some iteration $t$. A correct server $i$ should include $g_t^{(j)}$ while updating its model $\theta_t^{(i)}$, given network synchrony. The worker then does two steps in parallel: (1) estimates the updated model locally based on its own gradient: $\theta_{t+1}^{(j(l))}$ and (2) pulls a model $\theta_{t+1}^{(i)}$ from a parameter server $i$. If server $i$ is correct then, the growth of the pulled model $\theta_{t+1}^{(i)}$ (with respect to gradients) should be close to that of the estimated local model $\theta_{t+1}^{(j(l))}$, based on the guarantees given by the used GAR. Such a growth rate is encapsulated in the *Lipschitz coefficient*. If the pulled model is correct then, the worker expects that the Lipschitz coefficient computed based on that model is close to those of the other correct models received before by the worker. Concretely, a worker computes an empirical estimation of the Lipschitz coefficient $k = \left\| g_{t+1}^{(j)} - g_t^{(j)} \right\| / \left\| \theta_{t+1}^{(j(l))} - \theta_t^{(j)} \right\|$ and then, ensures that it follows the condition $k \leq K_p \triangleq \text{quantile}_{\frac{n_{ps} - f_{ps}}{n_{ps}}} \{K\}$, where $K$ is the list of all previous Lipschitz coefficients $k$ (i.e., with $t_{prev} < t$).

Note that such a filtering technique requires $n_{ps} > 3f_{ps}$ as Byzantine machines can craft models with Lipschitz coefficients that are deliberately placed in the first $2f_{ps}$ places of the set $K$. Hence, the tradeoff here is between the communication overhead and the required number of parameter server replicas: one can use robust aggregation of models, which requires only $n_{ps} > 2f_{ps}$, yet requires communicating with all servers in each round. In our design, we strive for reducing the communication overhead as much as possible, given communication is the bottleneck (36; 19).

**Models filter.** Although the Lipschitz filter can bound the model growth with respect to gradients, a server can trick this filter by sending a well-crafted model that is arbitrarily far from the other correct models. To overcome this problem, LIUBEI uses another filter, which we call *models filter*, to bound the distance between models in any two successive iterations. We assume all correct machines initialize models with the same state. Building upon the guarantees given by the used GAR, at iteration $t$, a worker can estimate an upper bound on the distance between two successive states of a correct model. Mathematically, the distance between a local estimate of a model $\theta_{t+1}^{(j(l))}$ and a pulled model $\theta_{t+1}^{(i)}$ is upper–bounded as follows:

$$\left\| \theta_{t+1}^{(j(l))} - \theta_{t+1}^{(i)} \right\| < \gamma_{T \cdot (t \mod T)} \left\| g_{T \cdot (t \mod T)} \right\| \left( (3T+2)(n_w - f_w)/4f_w + 2\big((t-1) \mod T\big) \right),$$

with $T = 1/3l\gamma_1$, where $l$ is the Lipschitz coefficient and $\gamma_t$ is the learning rate at iteration $t$. Such a bound is also based on the *scatter/gather* scheme we are using. The details of deriving this term is in the supplementary material, Section 3.4.2.

## 3.2 ALGORITHM

LIUBEI's algorithm operates in two phases: *scatter* and *gather*. One *gather* step is executed every $T$ iterations (line 8 to 11 in Algorithms 1 and 2); we call the whole $T$ iterations a *scatter* step.

Algorithms 1 and 2 depict the training loop applied by workers and servers respectively. As an initialization step, all machines in the network, be they workers or servers, initialize the model with the same random values, i.e., using the same *seed*. Empirical results show that this step is crucial in achieving high accuracy. Moreover, each worker $j$ chooses some random integer $r_j$ with $1 \leq r_j \leq n_{ps}$. Then each worker does a backpropagation step to compute its gradient estimate at the initial model. The subsequent steps $t \in \mathbb{N}$ work as follows.

The algorithm starts with a *scatter* step, which includes doing a few iterations. In each iteration, each parameter server $i$ pulls gradients $g_t$ from *all* workers and aggregates them using *MDA*, computing $g_{\text{agg}}^{(i)}$. Then, each server uses its own computed $g_{\text{agg}}^{(i)}$ to update the model. While parameter servers

are doing such computation, each worker $j$ does a speculative step by computing a local view of the updated model using its local computed gradient and its latest local model.

Then, each worker $j$ pulls one parameter vector from server $i$ where, $i = (r_j + t + 1) \mod n_{ps}$. Such a worker does a backpropagation step, computing the new gradient based on the pulled model. Based on this computation and the local estimate of the updated model, the worker applies the *Lipschitz filter* and the *models filter* to check the legitimacy of the pulled model. If the model fails to pass the filters, the worker $j$ pulls a new model from the parameter server $i$, where $i = (r_j + t + 2) \mod n_{ps}$. This process is repeated until a pulled model passes both filters.

To bound the drifts between parameter vectors at correct servers, every $T$ iterations, a global *gather* step is executed on both servers and workers sides: Each server $i$ sends to all other servers its current view of the model $\theta_t^{(i)}$. After gathering models from all servers, each server $i$ aggregates the received models using *Mean Around Median* (*MeaMed*)(34), computing $\theta_t^{(i(agg))}$. Then, each worker $j$ pulls such models $\theta_t^{(i(agg))}$ from all servers and aggregates them using *MeaMed*, before starting a new *scatter* step.

The interplay between using *MDA* to tolerate Byzantine workers and filters to tolerate Byzantine servers results in a convergence rate of $\mathcal{O}(\sqrt{(n_w - f_w)/n_w T^2})$. The formal proof of LIUBEI's Byzantine resilience is provided in the supplementary material (Section 3).

---

**Algorithm 1** Worker Algorithm

1: Calculate the value of $T$ and a value for *seed*
2: model ← init_model(seed)
3: r ← random_int(1,$n_{ps}$)
4: $t \leftarrow 0$
5: grad ← model.backprop()
6: **repeat**
7:     local_model ← apply_grad(model,grad)
8:     **if** $t \mod T = 0$ **then**
9:         models ← read_models()
10:         model ← MeaMed(models)
11:     **else**
12:         $i \leftarrow 0$
13:         **repeat**
14:             new_model ← read_model(
                $(r + t + i) \mod n_{ps}$)
15:             new_grad ← new_model.backprop()
16:             $i \leftarrow i + 1$
17:         **until** pass_filters(new_model)
18:         model ← new_model
19:         grad ← new_grad
20:     **end if**
21:     $t \leftarrow t + 1$
22: **until** $t >$ num_iterations

---

# 4   IMPLEMENTATION AND EVALUATION

## 4.1   IMPLEMENTATION

We use TensorFlow (5), a popular ML framework, as an underlying system for our implementation of LIUBEI. We use it only as a local library to *compute* gradients (on the workers' side) and *update* the model (on the parameter server side). In this sense, we do not rely on the *shared graph* design followed by TensorFlow. Yet, we allow each machine in the network, be it a server or a worker, to build its own independent graph. This gives better control over communication and disallows access of Byzantine machines to correct machines' memory (13). Moreover, this modular design allows LIUBEI to be integrated easily with other ML frameworks, e.g., PyTorch (28).

As a consequence of our design, we build our own communication abstractions. We use gRPC for communication and Protocol Buffers for serializing and deserializing data. Each machine in the network creates a gRPC server that serves requests coming from other machines. Such a request could be either asking for a gradient (usually from a server) or asking for an updated model (either from a worker or a server). We parallelize requests to multiple machines (for example when a server asks for gradients from all workers).

---

**Algorithm 2** Parameter Server Algorithm

1: Calculate the value of $T$ and a value for *seed*
2: model ← init_model(seed)
3: $t \leftarrow 0$
4: **repeat**
5:     grads ← read_gradients()
6:     grad ← MDA(grads)
7:     model.update(grad)
8:     **if** $t \mod T = 0$ **then**
9:         model ← read_models()
10:         model ← MeaMed(models)
11:     **end if**
12:     $t \leftarrow t + 1$
13: **until** $t >$ num_iterations

---

## 4.2   SETUP

**Baselines.** We compare LIUBEI to two baselines: TensorFlow (5) and GuanYu (15). The first baseline shows the performance of training using a non–Byzantine resilient distributed framework and hence, comparing LIUBEI to it quantifies the cost of Byzantine resilience. GuanYu is (so far) the only algorithm that tolerates both Byzantine servers and workers. Comparing LIUBEI to GuanYu highlights the performance gain achieved by our algorithm.

**Metrics.** We use two standard metrics for evaluating LIUBEI: Accuracy and Throughput. *Accuracy* denotes the fraction of correct answers a model gives (correct classifications) among all its answers, using the *test* set. We show the progress of accuracy with the number of iterations and time. *Throughput* measures the number of updates the system, i.e., the parameter server does per second.

**Testbed.** Our experimental platform is Grid5000 (2). We employ nodes, each having 2 CPUs (Intel Xeon E5-2630 v4) with 10 cores, 256 GiB RAM and $2\times10$ Gbps Ethernet. Unless otherwise stated, we employ 20 compute nodes (workers), out of them (up to) 8 nodes could be Byzantine. In the case of vanilla TensorFlow deployment, we use only 1 machine as a parameter server. Otherwise, we employ 4 machines for LIUBEI deployment and 5 machines for GuanYu deployment; these numbers are to tolerate at most 1 Byzantine server, based on the requirements of each algorithm.

**Dataset and Model.** We consider the image classification task due to its wide adoption as a benchmark for distributed ML systems, e.g., (12). We use MNIST (3) and CIFAR-10 (1) datasets. MNIST is a dataset of handwritten digits. It has 70,000 $28 \times 28$ images in 10 classes. CIFAR-10 is a widely–used dataset in image classification (33; 36). It consists of 60,000 $32 \times 32$ colour images in 10 classes.

Table 1: Models used to evaluate LIUBEI.

| Model | # parameters | Size (MB) |
|---|---|---|
| MNIST_CNN | 79510 | 0.3 |
| CifarNet | 1756426 | 6.7 |
| Inception | 5602874 | 21.4 |
| ResNet-50 | 23539850 | 89.8 |
| ResNet-200 | 62697610 | 239.2 |

We employ different models with different sizes ranging from simple models like small convolutional neural network (CNN) for MNIST, training a few thousands of parameters to big models like ResNet-200 with around 63M parameters. All models are listed in Table 1.

### 4.3 RESULTS

We show here the key results of the evaluation of LIUBEI. First, we show the progress of accuracy over training iterations and time (i.e., convergence) and then, we discuss the throughput of LIUBEI compared to GuanYu (15); both sets of experiments are done in a Byzantine–free environment. Then, we show the performance of LIUBEI in a Byzantine environment, i.e., under a recent attack. Finally, we describe the effect of changing the value of $T$ on the filters' performance and convergence.

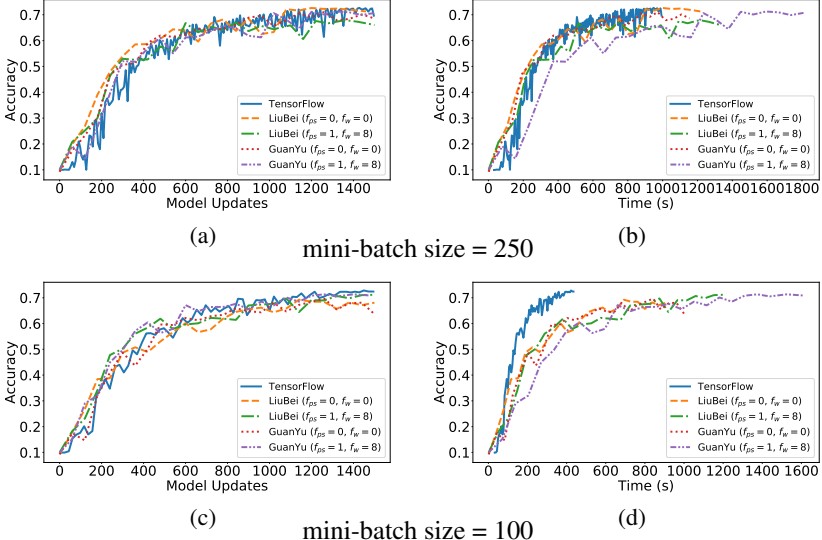

Figure 2: Convergence in a non-Byzantine environment.

**Convergence.** Figure 2 shows the convergence of all experimented systems with both time and training iterations. We experiment with two batch sizes and two values for declared Byzantine servers and workers (only for the Byzantine–tolerant deployments). Figure 2a shows that all deployments have almost the same convergence, with a slight loss in final accuracy for the Byzantine–tolerant deployments, which we quantify to around 5%. Such a loss is emphasized with the smaller batch size (Figure 2c). This accuracy loss is admitted in previous work (34; 15) and inherits from using

statistical methods (basically, GARs) for Byzantine resilience. Such GARs ensure convergence only to a ball around the optimal solution, i.e., local minimum (8; 16). As LIUBEI builds on these GARs, we expect it to achieve an accuracy similar to that achieved by such GARs. Moreover, the figures confirm that using a higher batch size gives a better performance for both LIUBEI and GuanYu. Figures 2a and 2c show that LIUBEI achieves the same convergence behavior as GuanYu.

The cost of Byzantine resilience is more clear when convergence is observed over time (Figure 2b), especially with the lower batch size (Figure 2d). Using a bigger batch size helps the Byzantine–tolerant systems achieve a performance close to vanilla TensorFlow. We quantify the overhead of LIUBEI compared to vanilla TensorFlow to around 24% and the performance gain of LIUBEI compared to GuanYu by around 70%. We analyze this overhead with more scenarios later on in the section. We draw two main observations from these figures: First, unlike with GuanYu, changing the number of declared Byzantine machines does not affect the progress of accuracy while deploying LIUBEI. This is because servers and workers in LIUBEI deployments always wait for replies from all machines, regardless of the number of Byzantine machines (unlike in GuanYu, where the number of expected replies depends on the number of Byzantine machines). Second, LIUBEI always outperforms GuanYu, especially with non–zero values for declared Byzantine servers and workers. Such a result is expected as LIUBEI uses less number of communication rounds and less number of messages per round compared to GuanYu. Given that distributed ML systems are network–bound (19; 36), reducing the communication overhead significantly boosts the performance and the scalability of such systems.

**Throughput.** We do the same experiment again, yet with different state–of–the–art models so as to quantify the through-put gain of LIUBEI compared to GuanYu. Figure 3 shows the throughput of LIUBEI divided by the throughput of GuanYu in each case. For the current version of our implementation, we use CPUs for computation; we expect that using GPUs will emphasize more the effectiveness of our LIUBEI protocol as the communication overhead is generally highlighted more with GPUs. From this figure, we see that LIUBEI outperforms GuanYu in all cases, where the performance gain is emphasized

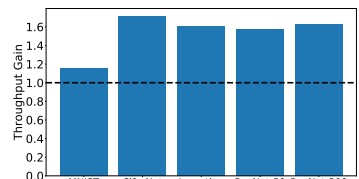

Figure 3: Throughput gain of LI-UBEI compared to GuanYu (15).

more with large models. This is expected because the main advantage of LIUBEI is to decrease the number of communication messages, where bigger messages are transmitted with bigger models.

**Byzantine workers.** We study here the performance of LIUBEI in the presence of Byzantine workers. Simple misbehavior like message drops, unresponsive machines or reversed gradients are well-studied and have been shown to be tolerated by Byzantine–resilient GARs (34; 15), which LIUBEI also uses. Thus, here we focus on a more recent attack that is coined as *A little is enough attack* (6). Such an attack states that changing each dimension in gradients of Byzantine machines, even by a very small value, can trick some of Byzantine–resilient GARs like (8; 16).

We apply this attack to multiple deployments of LIUBEI. In each scenario, we apply the strongest possible change in gradients' coordinates so as to hamper the convergence the most. We study the effect of this attack on the convergence of LIUBEI with both the ratio of Byzantine machines to the total number of machines (Figure 4a) and the batch size (Figure 4b). We use the deployment with no declared Byzantine behavior (that is presented in Figure 2a) as a baseline in this experiment.

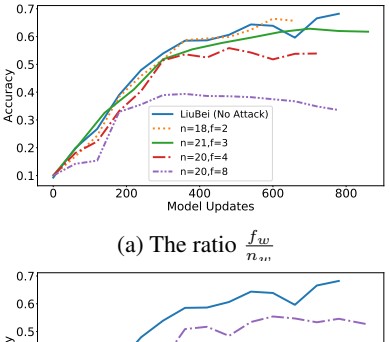

(a) The ratio $\frac{f_w}{n_w}$

(b) Batch size

Figure 4: Convergence in the presence of Byzantine workers.

Figure 4a shows that the effect of the attack starts to appear clearly when the number of Byzantine nodes is a

significant fraction (more than 20%) of the total number of nodes. This is intuitive as the attack tries to increase the variance between the submitted gradients to the parameter servers and hence, increases the ball (around the local minimum) to which the used GAR converges (see e.g. (8; 16) for a theoretical analysis of the interplay between the variance and the Byzantine resilience). Stretching

the number of Byzantine machines to the maximum allowed ($f_w = 8$) downgrades the accuracy to around 40% (compared to 67% in "No Attack" case). The main reason for this is that the assumption on the variance between gradients required by *MDA* (the used GAR in this deployment) is not satisfied. We discuss this issue in details in the supplementary material (Section 4).

Increasing the batch size not only improves the accuracy per training iteration but also the robustness of LIUBEI (by narrowing down the radius of the ball around the convergence point, where the model will fluctuate as proven in (8; 16)). Figure 4b fixes the ratio of $f_w$ to $n_w$ to the biggest allowed value to see the effect of using a bigger batch size on the convergence behavior. This figure confirms that increasing the batch size increases the robustness of LIUBEI. Moreover, based on our experiments, setting 25% of nodes to be Byzantine while using a batch size of (up to) 256 does not experimentally satisfy the assumption on the variance of *MDA* in this deployment, which leads to a lower accuracy after convergence (see supplementary material, Section 4).

**Byzantine servers.** Figure 5 shows the performance of LIUBEI in the presence of 1 Byzantine server out of a total of 4 servers. We experimented with 4 Byzantine behavior: **1.** *Reversed*, in which the server sends a correct model multiplied by a negative number, **2.** *Partial Drop*, in which the server randomly chooses 10% of the weights and set them to zero (this simulates using unreliable transport protocol in the communication layer, which was proven beneficial, e.g., (13)), **3.** *Random*, in which the server replaces the learned weights by random numbers, and **4.** *LIE*, an attack inspired from the *little is enough* attack (6),

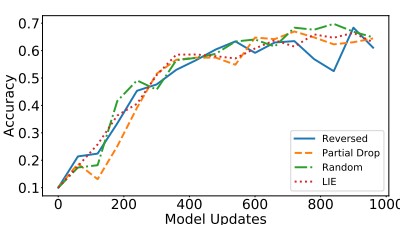

Figure 5: Convergence in the presence of a Byzantine server.

in which the server multiplies each of the individual weights by a small number $z$, where $|z - 1| < \delta$ with $\delta$ very close to zero; $z = 1.035$ in our experiments. Such a figure shows that LIUBEI can tolerate the experimented Byzantine behavior and that the learning converges to the same accuracy observed before in Figure 2a. We noted that both filters together do not pass any false positives, i.e., the models submitted from the Byzantine server never pass both filters (although could trick one filter individually).

**Effect of changing $T$.** The value of $T$ denotes the number of iterations done in one *scatter* step (i.e., before executing one *gather* step). Figure 6 shows the effect of changing the value of $T$ on convergence with both time and model updates in a Byzantine-free environment. Figure 6a shows that the value of $T$ almost does not have any effect on the convergence w.r.t. the model updates. This happens because models on correct servers almost do not drift from each other (as all the servers are correct). Interestingly, Figure 6b shows that using a higher value for $T$ helps converge faster. This is because increasing $T$ decreases the communication overhead, achieving faster updates and higher throughput. However, it is important to note that as the value of $T$ increases, the expected drifts between models on correct servers increases, and it becomes easier for the Byzantine server to trick the workers.

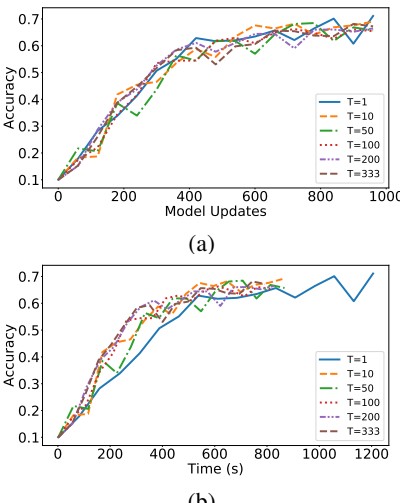

Figure 6: The effect of changing $T$ on performance.

Figure 7 shows the ratio of false negatives (i.e., the number of falsely-rejected correct models by filters on the workers) to the total number of submitted models, with different values for $T$. Although we declare $f_{ps} = 1$, we do not employ any Byzantine behavior in this experiment. We observe the number of rejected models on workers after 500 learning iterations. In general, the ratio of false negatives never exceeds 1% in this experiment, and it is almost stable with increasing $T$. Note that 333 is the maximum value allowed for $T$ in this setup (to follow the safety rules of LIUBEI). With $T = 1$, the false negatives are always zero, simply because the filters do not work in this setup (i.e., the *gather* step is executed in every iteration). Such a figure shows that our filtering mechanism is effective in not producing many false negatives and hence, do not waste communication rounds (when a model is rejected, the worker asks for another model from a different server).

## 5   RELATED WORK

The closest approach to ours is GuanYu (15), which toler­ates Byzantine workers and servers, considering however an asynchronous network. GuanYu uses a GAR to toler­ate Byzantine workers and *Median* (34) twice in a row in each iteration at both workers and parameter servers to tolerate Byzantine servers. We note two limitations in this work. First, GuanYu's design induces an overhead both in computation and communication: the algorithm requires three communication rounds and applies aggregation rules three times in each iteration. In addition, it requires more compute nodes than the vanilla case to tolerate network

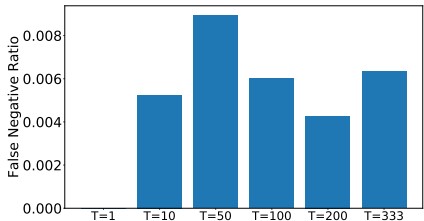

Figure 7: False negative percentage with different values of $T$.

asynchrony. Such an overhead limits scalability for large–scale applications that use models with millions of parameters. Second, for GuanYu to work, the authors assume that the difference between models on correct parameter servers is always bounded; this cannot be fulfilled in some cases. We address these two problems by using a new filtering mechanism instead of models aggregation and by employing a *gather* step every few iterations accordingly.

Several proposals tolerate only Byzantine workers in synchronous/asynchronous network and syn­chronous/asynchronous learning, all assuming a single correct parameter server. (34) proposed three Median-based aggregation rules that can resist both Byzantine and Dimensional attacks. Krum (8) and Multi-Krum (13) use a distance-based algorithm to filter out the Byzantine inputs and average the correct ones. Bulyan (16) proposes a meta-algorithm that guarantees a strong Byzantine resilience, i.e., resist a strong adversary that can fool the aforementioned algorithms. Draco (11) uses coding schemes and redundant gradient computation for Byzantine resilience, where Detox (29) combines coding schemes with robust aggregation for better resilience and overhead guarantees. Kardam (14) uses filtering techniques to filter out Byzantine workers in asynchronous learning setup.

## 6   CONCLUDING REMARKS

**Summary.**   This paper proposed LIUBEI, a robust Machine Learning (ML) algorithm that does not trust any individual component in the network almost without adding communication rounds compared to vanilla distributed ML systems. LIUBEI relies on the *server/worker* architecture and assumes network synchrony. LIUBEI introduces a novel filtering mechanism used by workers to check whether a pulled model (from a parameter server) is suspicious or not, without requiring to pull models from all servers. LIUBEI also introduces a novel protocol, *scatter/gather*, to reduce the communication overhead. We theoretically prove that LIUBEI tolerates Byzantine servers and workers and ensures convergence. Our experiments with a TensorFlow–based implementation show that LIUBEI's overhead, in terms of convergence, is around 24% compared to vanilla TensorFlow, where its throughput gain is around 70% compared to a state–of–the–art solution that tolerates Byzantine workers and servers in asynchronous networks.

**Open question.**   The relation between the frequency of applying the *gather* step and the variance between models on correct servers is data–dependent and model–dependent. In our analysis, we provide safety guarantees on this relation that will always ensure Byzantine resilience and convergence. However, we believe that in some cases applying the *gather* step more frequently may lead to a noticeable improvement in the convergence speed as it decreases the variance. The trade-off between this gain and the overhead of communication (to gather) remains a data-dependent and model-dependent question that is out of the scope of this work.

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

# Fast Machine Learning with
# Byzantine Workers and Servers

## Supplementary Material

## Quick Note

We give here the convergence proof of LiuBei along with its resilience guarantees. LiuBei addresses the total Byzantine resilience problem in distributed Machine Learning (ML) context (following the parameter server architecture [8]) where, both workers and servers could behave arbitrarily. LiuBei assumes the network to be synchronous, i.e., there is an upper bound (not necessarily known) on the computation and communication delay. LiuBei uses a Byzantine–resilient gradient aggregation rule (GAR) to tolerate Byzantine workers, which is *Minimum–Diameter Averaging* [9] (or in short *MDA*). Such a GAR can be safely replaced with any other GAR that gives similar guarantees given by the used GAR such as Krum [2] or Median [11]. The main goal of LiuBei is to reduce communication overhead as much as possible, while guaranteeing resilience against Byzantine workers and servers. This is mainly achieved by our novel filtering technique and our *scatter/gather* protocol.

**Note about the name.** *Lui Bei* was the warlord of *Guan Yu*, the general whose name is given to the first algorithm to tolerate both Byzantine workers and servers. We show that LiuBei performs better than *GuanYu*.

## 1 Preliminary Material

### 1.1 Robust aggregation

Robust aggregation of gradients is the key for Byzantine workers' resilience. To this end, gradients are processed by a gradient aggregation rule (GAR), which purpose is to ensure that output of aggregation is as close as possible to the real gradient of the loss function.
In the general theory of stochastic gradient descent (SGD) convergence, a typical validity assumption is that the gradient estimator is unbiased [3]. The role of a GAR is to ensure a relaxed version of this assumption in order to accommodate for the presence of malicious workers (whose gradients are potentially biased).
Definition 1 gives such a relaxation, which we adapt from [2, 7] and which was used as a standard for Byzantine resilience in, e.g. [12, 13, 11].

**Definition 1.** *Let $\alpha \leq 0 < \pi/2$ be any angular value and $0 \leq f \leq n$ with $n$ the total number of input vectors to the GAR and $f$ the maximum number of Byzantine vectors. Let $g$ be an unbiased estimate of the true gradient $G$, i.e., $\mathbb{E}\, G = g$.*
*A GAR (which output noted as $\mathcal{F}$) is robust (said to be $(\alpha, f)$–Byzanitne resilient) if*
$$\langle \mathbb{E}\, \mathcal{F}, g \rangle \geq (1 - \sin \alpha). \|g\|^2 > 0$$

LiuBei uses two GARs: *Minimum–Diameter Averaging* [7] and *Mean Around Median* [11].

## 1.2 *Minimum–Diameter Averaging* (*MDA*)

*MDA* is a gradient aggregation rule (GAR) that ensures resilience against a minority of Byzantine input gradients. Mathematically, this function was introduced in [9] and its Byzantine resilience proof was given in [7]. *MDA* satisfies the $(\alpha, f)$ Byzantine resilience guarantees[1] introduced in [2]. Formally, let $\mathcal{X}$ be the set of input gradients (all follow the same distribution), out of them $f$ are Byzantine, and $y$ be the output of the Byzantine resilient GAR. Then, the following properties hold:

1. $\mathbb{E}\,y$ is in the same half–space as $\mathbb{E}\,\mathcal{X}$.
2. the first 4 statistical moments of $y$ are bounded above by a linear combination of the first 4 statistical moments of $x \sim \mathcal{X}$.

Such conditions are sufficient to show that the output of this GAR guarantees convergence of the learning procedure. More formally, these conditions enable the GAR to have a proof that follows from the *global confinement* proof of Stochastic Gradient Descent (SGD) [3].

In order to work, *MDA* assumes the following (as any other GAR):

$$\exists \kappa \in \,]1, +\infty[\,, \ \forall (i, t, \theta) \in [1 \mathinner{..} n - f] \times \mathbb{N} \times \mathbb{R}^d, \ \kappa \frac{2f}{n-f} \sqrt{\mathbb{E}\left(\left\|g_t^{(i)} - \mathbb{E} g_t^{(i)}\right\|^2\right)} \leq \|\nabla L\left(\theta\right)\|, \quad (1)$$

where $\theta$ is the model state at the training iteration $t$, $n$ is the total number of input gradients, $f$ is the maximum number of Byzantine gradients, $g_t$ is an unbiased estimate of the gradient at iteration $t$, and $L$ is the loss function.

The *MDA* function works as follows: Consider that the total number of gradients is $n$ and the maximum number of Byzantine gradients is $f$ with $n \geq 2f + 1$. *MDA* enumerates all subsets of size $n - f$ from the input gradients and finds the subset with the *minimum diameter* among all subsets of this size, i.e., $n - f$. The *diameter* of a subset is defined as the maximum distance between any two elements of this subset. The output of the *MDA* function is the average of gradients in such a subset. More formally, the *MDA* function is defined as follows [7]:

Let $(g_1 \ldots g_n) \in \left(\mathbb{R}^d\right)^n$, and $\mathcal{X} \triangleq \{\,g_1 \ldots g_n\,\}$ the set containing all the input gradients.

Let $\mathcal{R} \triangleq \{\, \mathcal{Q} \mid \mathcal{Q} \subset \mathcal{X}, \ |\mathcal{Q}| = n - f \,\}$ the set of all the subsets of $\mathcal{X}$ with a cardinality of $n - f$, and let $\mathcal{S} \triangleq \underset{\mathcal{Q} \in \mathcal{R}}{\arg\min} \left(\underset{(g_i, g_j) \in \mathcal{Q}^2}{\max} \left(\|g_i - g_j\|\right)\right)$.

Then, the aggregated gradient is given by $MDA\left(g_1 \ldots g_n\right) \triangleq \frac{1}{n-f} \sum\limits_{g \in \mathcal{S}} g$.

## 1.3 *Mean Around Median* (*MeaMed*)

The *MeaMed* function is a fairly intuitive function to remove outliers from an input set. As an aggregation rule, *MeaMed* was proven to be $(\alpha, f)$ Byzantine resilient in [11]. Even more than that, *MeaMed* can resist dimensional attacks, i.e., dimensional $(\alpha, f)$ Byzantine resilient. In this paper, we apply *MeaMed* to models, rather than gradients not only to exclude Byzantine models but also to bring models on correct servers close to each other.

*MeaMed* works as follows: For each dimension (in $d$), *MeaMed* finds the *median* value among all input models and then averages the closest $n - f - 1$ values to this value, along with the median value. Formally, the *MeaMed* function is defined as follows:

Let $\mathcal{X}$ be the set of input models of size $n$, where the maximum number of Byzantine models is $f$ with $n \geq 2f + 1$.

Let the output of *MeaMed* be $\rho$, with $\rho_j$ is the dimension/coordinate $j \in [1 \ldots d]$ in the output model.

---

[1]Basically, any GAR that satisfies such a form of resilience [2, 7, 11] can be used with LIUBEI; *MDA* is just an instance.

Let $m_j$ be the median value of all values in dimension $j$ at all input models $\theta$ in $\mathcal{X}$.

Then, $\rho_j \triangleq \frac{1}{n-f} \sum_{\theta_j \in \mathcal{S}_j} \theta_j$, where $\mathcal{S}_j$ is the set of closest values to $m_j$ in dimension $j$ among all inputs, including $m_j$, i.e., $|S_j| = n - f$.

# 2 LiuBei's Algorithm

## 2.1 Notations

Let $(n_w, n_{ps}, f_w, f_{ps}, d) \in \mathbb{N}^5$, each representing:
- $n_{ps} \geq 3f_{ps} + 1$ the total number of parameter servers, among which $f_{ps}$ are Byzantine
- $n_w \geq 2f_w + 1$ the total number of workers, among which $f_w$ are Byzantine
- $d$ the dimension of the parameter space $\mathbb{R}^d$

Let (without loss of generality):
- $[1 .. n_{ps} - f_{ps}]$ be indexes of correct parameter servers
  $[n_{ps} - f_{ps} + 1 .. n_{ps}]$ be indexes of Byzantine parameter servers
- $[1 .. n_w - f_w]$ be indexes of correct workers
  $[n_w - f_w + 1 .. n_w]$ be indexes of Byzantine workers

Let $\theta_t^{(i)}$ be a notation for the parameter vector (i.e., model) at parameter server $i \in [1 .. n_{ps} - f_{ps}]$ for step $t \in \mathbb{N}$.

Let $g_t^{(i)}$ be a notation for the gradient estimation at worker $i \in [1 .. n_w - f_w]$ for step $t \in \mathbb{N}$.

Let $G_t^{(i)}$ be a notation for the gradient distribution at worker $i \in [1 .. n_w - f_w]$ for step $t \in \mathbb{N}$.

Let $L$ be the loss function we aim to minimize, let $\nabla L(\theta)$ be the real gradient of the loss function at $\theta$, and let $\widehat{\nabla L}(\theta)$ be a stochastic estimation of the gradient, following $G$, of $L$ at $\theta$.

Let $\gamma_t$ be the learning rate at the learning iteration $t \in \mathbb{N}$ with the following specifications:

1. The sequence of learning rates $\gamma_t$ is decreasing [2] with $t$, i.e., if $t_a > t_b$ then, $\gamma_{t_a} < \gamma_{t_b}$. Thus, the initial learning rate $\gamma_0$ is the largest value among learning rates used in subsequent steps.

2. The sequence of learning rates $\gamma_t$ satisfies $\sum_t \gamma_t = \infty$ and $\sum_t \gamma_t^2 < \infty$.

## 2.2 Algorithm

**Initialization.** Each correct parameter server $i$ and worker $j$ starts (at step $t = 0$) with the same parameter vector:

$$\forall i \in [1 .. n_{ps} - f_{ps}], \ \theta_0^{(i)} \triangleq \theta_0$$

$$\forall j \in [1 .. n_w - f_w], \ \theta_0^{(j)} \triangleq \theta_0$$

Each correct worker $j$ generates a random integer $r_j \in [1 .. n_{ps}]$ and does one backpropagation step to compute $g_0^{(j)}$ at the initial model $\theta_0$.

**Training loop.** Each training step $t \in \mathbb{N}$, the following sub-steps are executed sequentially (unless otherwise stated).

1. Each parameter server $i$ pulls gradients $g_t$ from *all* workers and then applies the *MDA* function on the received gradients, computing the aggregated gradient $g_{\text{agg}}^{(i)}$. Then, each server uses its own computed $g_{\text{agg}}^{(i)}$ to update the model as follows: $\theta_{t+1}^{(i)} = \theta_t^{(i)} - \gamma_t g_{\text{agg}}^{(i)}$.

2. While parameter servers are doing step 1, each worker $j$ does a speculative step as follows: a worker $j$ calculates its local view to the updated model: $\theta_{t+1}^{(j(l))} = \theta_t^{(j)} - \gamma_t g_t^{(j)}$.

---

[2]In fact, it is sufficient that the sequence is decreasing only once every $T$ steps, with $T = \frac{1}{3.l.\gamma_1}$ where $l$ is the Lipschitz coefficient of assumption 5 (cf Section 3.1).

3. Each worker $j$ pulls one parameter vector $\theta_{t+1}^{(i)}$ from server $i$ where, $i = (r_j + t + 1) \mod n_{ps}$.

4. Each worker $j$ does the backpropagation step, computing $g_{t+1}^{(j)}$ at the pulled model $\theta_{t+1}^{(i)}$.

5. Each worker $j$ tests the legitimacy of the received model $\theta_{t+1}^{(i)}$ using the *Lipschitz criterion* (i.e., Lipschitz filter) and the *difference on model norms* (i.e., models filter) as follows. First, a worker $j$ calculates $k$, an empirical estimation of the Lipschitz coefficient, which is defined as:

$$k = \frac{\left\| g_{t+1}^{(j)} - g_t^{(j)} \right\|}{\left\| \theta_{t+1}^{(j(l))} - \theta_t^{(j)} \right\|}$$

Then, the worker tests whether this value $k$ lies in the non-Byzantine quantile of Lipschitz coefficients: $k \leq K_p \triangleq \text{quantile}_{\frac{n_{ps} - f_{ps}}{n_{ps}}} \{K\}$ where, $K$ is the list of all previous Lipschitz coefficients $k$ (i.e., with $t_{prev} < t$). Second, the worker $j$ computes the distance between the local and the pulled (from server $i$) models as follows: $\left\| \theta_{t+1}^{(j(l))} - \theta_{t+1}^{(i)} \right\|$ and makes sure that such a difference is $< \gamma_{T \cdot (t \mod T)} \left\| g_{T \cdot (t \mod T)} \right\| \left( \frac{(3T+2)(n_w - f_w)}{4 f_w} + 2 \left( (t-1) \mod T \right) \right)$, with $T = 1/3 l \gamma_1$, where $l$ is the Lipschitz coefficient (assumption 5). Such a difference is instructed by a correct execution of the *MDA* algorithm. If both conditions are satisfied, the received model $\theta_{t+1}^{(i)}$ is approved and the algorithm continues normally. Otherwise, the parameter server $i$ is suspected and its model is ignored; worker $j$ continues by repeating iteration $t$ again from step 3.

6. To bound the drifts between parameter vectors at correct servers, each $T = 1/3 l \gamma_1$ steps, a global *gather* step is executed on both servers and workers sides. This step is executed as follows: Each server $i$ sends to all other servers its current view of the model $\theta_t^{(i)}$. After gathering models from all servers, each server $i$ aggregates such models with *MeaMed*, computing $\theta_t^{(i(agg))}$. Then, each worker $j$ pulls the model $\theta_t^{(i(agg))}$ from all parameter servers and aggregates the received models using *MeaMed*. Finally, each worker $j$ uses the aggregated model to compute the backpropagation step, and the algorithm continues normally from step 1.

We call steps 1 through 5 **scatter** step and step 6 **gather** step. During *scatter* step(s) servers do not communicate and hence, their views of the model deviate from each other. The goal of the *gather* step is to bring back the models at the correct servers close to each other.

# 3 LiuBei's Convergence

In this section, we show that LiuBei guarantees convergence and tolerates Byzantine workers and servers. We first state the assumptions for LiuBei to work and then dive into the proof, which is shown step by step: First, we show that the result of execution of *MDA* on two correct servers results in models that are close to each other. Then, we show how we bound the distance between models on correct servers throughout the learning procedure using the *scatter/gather* protocol. After that, we explain our novel filtering mechanism and show why they can tolerate Byzantine servers. Finally, we put everything together to show the convergence of LiuBei.

## 3.1 Assumptions

1. $\forall t \in \mathbb{N}$, $g_t^{(1)} \ldots g_t^{(n_w - f_w)}$ are mutually independent.

2. $\exists \sigma' \in \mathbb{R}_+, \ \forall (i, t) \in [1 .. n_w - f_w] \times \mathbb{N}, \ \mathbb{E} \left\| g_t^{(i)} - \mathbb{E} g_t^{(i)} \right\| \leq \sigma'$.

3. $L$ is positive, and 3–times differentiable with continuous derivatives.

4. $\forall r \in [2 .. 4], \ \exists (A_r, B_r) \in \mathbb{R}^2, \ \forall (i, t, \theta) \in [1 .. n_w - f_w] \times \mathbb{N} \times \mathbb{R}^d, \ \mathbb{E} \left\| g_t^{(i)} \right\| \leq A_r + B_r \|\theta\|^r$.

5. $L$ is Lipschitz continuous, i.e. $\exists l > 0, \ \forall (x, y) \in \left(\mathbb{R}^d\right)^2, \ \|\nabla L(x) - \nabla L(y)\| \leq l \|x - y\|$.

6. $\exists D \in \mathbb{R}, \ \forall \theta \in \mathbb{R}^d, \ \|\theta\|^2 \geq D, \ \exists (\varepsilon, \beta) \in \mathbb{R}_+ \times [0, \pi/2 - \gamma[,$
   $\|\nabla L(\theta)\| \geq \varepsilon, \ \langle \theta, \nabla L(\theta) \rangle \geq \cos(\beta) \|\theta\| \|\nabla L(\theta)\|$.

Assumptions 1 to 5 (i.i.d, bounded variance, differentiability of the loss, bounded statistical moments, and Lipschitz continuity of the gradient) are the most common ones in classical SGD analysis [3, 4].

Assumption 6 was first adapted from [3] by [2, 7, 5] to account for Byzantine resilience. It intuitively says that beyond a certain horizon, the loss function is "steep enough" (lower bounded gradient) and "convex enough" (lower bounded angle between the gradient and the parameter vector). The loss function does not need to be convex, but adding regularization terms such as $\|\theta\|^2$ ensures assumption 6, since close to infinity, the regularization dominates the rest of the loss function and permits the gradient $\nabla L(\theta)$ to point to the same half space as $\theta$. The original assumption of [3] is that $\langle \theta, \nabla L(\theta) \rangle > 0$; in [2, 7] it was argued that requiring this scalar product to be strictly positive is the same as requiring the angle between $\theta$ and $\nabla L(\theta)$ to be lower bounded by an acute angle ($\beta < \pi/2$).

## 3.2 Bounded gradient aggregation on correct parameter servers

Consider two correct servers $x$ and $z$, this section discusses the maximum difference between aggregated gradients (using $MDA$) on both servers at any step $t \in \mathbb{N}$.

Let $(d, f_w) \in (\mathbb{N} - \{0\})^2$ and $n_w \geq 2f_w + 1$. Let $H_x$ be the set of indices of *correct* gradients received at server $x$ (and the same notation for $H_z$) and $g_i^{(j)}$ be the gradient received from worker $i$ at server $j$ (only for this section).

In the following, we show that:

$$\exists c \in \mathbb{R}_+, \ \forall \left(g_i^{(j)}\right) \in \left(\mathbb{R}^d\right)^{n_w . n_{ps}},$$
$$\left\| MDA \left(g_1^{(x)} \ldots g_{n_w}^{(x)}\right) - MDA \left(g_1^{(z)} \ldots g_{n_w}^{(z)}\right) \right\| \leq c \cdot \max_{(i,j) \in (H_x \cup H_z)^2} \|g_i - g_j\|$$

*Proof.* Based on the definition of the $MDA$ function (Section 1.2), let $\mathcal{S}$ be the set of the chosen gradients by the $MDA$ algorithm, i.e., with the minimum diameter among all subsets of size $n_w - f_w$. Then, the aggregated gradient is given by $MDA(g_1 \ldots g_{n_w}) \triangleq \frac{1}{n_w - f_w} \sum_{g \in \mathcal{S}} g$.

Based on this definition, the following holds:

$$\exists (i, j) \in H_x^2, \ \forall (y, v) \in \mathcal{S}^2, \ \|y - v\| \leq \|g_i - g_j\|.$$

Then, observing that $n_w - f_w > f_w \implies \exists k \in H_x, \ g_k \in \mathcal{S}$. Using the triangle inequality, we

have:

$$\|MDA\,(g_1 \ldots g_{n_w}) - g_k\| = \left\|\left(\frac{1}{n_w - f_w}\sum_{g \in \mathcal{S}} g\right) - g_k\right\|$$

$$= \frac{1}{n_w - f_w}\left\|\sum_{g \in \mathcal{S}}(g - g_k)\right\|$$

$$\leq \frac{1}{n_w - f_w}\sum_{g \in \mathcal{S}}\|g - g_k\|$$

$$\leq \frac{1}{n_w - f_w}\sum_{g \in \mathcal{S}}\left(\max_{(i,j) \in H_x{}^2}\|g_i - g_j\|\right)$$

$$\leq \max_{(i,j) \in H_x{}^2}\|g_i - g_j\|.$$

Based on this, we have

$$\left\|MDA\left(g_1^{(x)} \ldots g_{n_w}^{(x)}\right) - MDA\left(g_1^{(z)} \ldots g_{n_w}^{(z)}\right)\right\|$$

$$= \left\|MDA\left(g_1^{(x)} \ldots g_{n_w}^{(x)}\right) - g_k + g_k - g_l + g_l - MDA\left(g_1^{(z)} \ldots g_{n_w}^{(z)}\right)\right\|$$

$$\leq \left\|MDA\left(g_1^{(x)} \ldots g_{n_w}^{(x)}\right) - g_k\right\| + \|g_k - g_l\| + \left\|MDA\left(g_1^{(z)} \ldots g_{n_w}^{(z)}\right) - g_l\right\|$$

$$\leq 3 \cdot \max_{(i,j) \in (H_x \cup H_z)^2}\|g_i - g_j\|$$

Thus,

$$\left\|MDA\left(g_1^{(x)} \ldots g_{n_w}^{(x)}\right) - MDA\left(g_1^{(z)} \ldots g_{n_w}^{(z)}\right)\right\| \leq 3 \cdot \max_{(i,j) \in (H_x \cup H_z)^2}\|g_i - g_j\| \qquad (2)$$

$\square$

In plain text, this equation bounds the difference between aggregated gradients at two different correct servers, based on the maximum distance between two correct gradients, i.e., aggregated gradients on different servers will not drift arbitrarily.

## 3.3 Bounded distance between correct models

To satisfy Equation 1 (the required assumption by *MDA*, Section 1.2), models at correct parameter servers should not go arbitrarily far from each other. Thus, a global *gather* step (step 6 in Section 2.2) is executed once in a while to bring the correct models back close to each other. Such a gather step is close to the distributed contraction module of [6], which is based on applying *Median* on both servers and workers sides (with the difference that the *gather* step applies *MeaMed* instead of *Median* and does not execute in all iterations).

In this section we quantify the maximum number of iterations that can be executed in one *scatter* step before executing one *gather* step. From another perspective, the goal is to find the maximum possible distance between correct models that still satisfies the requirement of *MDA* on the distance between correct gradients (Equation 1).

Without loss of generality, assume two correct parameter servers $x$ and $z$ starting with the same initial model $\theta_0$. After the first iteration, their updated models are given by:

$$\theta_1^{(x)} = \theta_0 - \gamma_1 MDA\left(g_1^{(1)} \ldots g_1^{(n_w)}\right)_x$$

$$\theta_1^{(z)} = \theta_0 - \gamma_1 MDA\left(g_1^{(1)} \ldots g_1^{(n_w)}\right)_z$$

Thus, the difference between them is given by:

$$\left\|\theta_1^{(x)} - \theta_1^{(z)}\right\| = \gamma_1 \left\|MDA\left(g_1^{(1)}\ldots g_1^{(n_w)}\right)_z - MDA\left(g_1^{(1)}\ldots g_1^{(n_w)}\right)_x\right\|$$

In a perfect environment, with no Byzantine workers, this difference is zero, since the input gradients to the $MDA$ function at both servers are the same (no worker lies about its gradient estimation), and the $MDA$ function is deterministic (i.e., the output of $MDA$ computation on both servers is the same). However, a Byzantine worker can send different gradients to different servers while crafting these gradients carefully to trick the $MDA$ function to include them in the aggregated gradient (i.e., force $MDA$ to select the malicious gradients in the set $\mathcal{S}$). In this case, $\left\|\theta_1^{(x)} - \theta_1^{(z)}\right\|$ is not guaranteed to be zero. Based on Equation 2, the difference between the result of applying $MDA$ in the same iteration is bounded and hence, such a difference can be given by:

$$\left\|\theta_1^{(x)} - \theta_1^{(z)}\right\| \leq 3 \cdot \gamma_1 \cdot \max_{(i,j)\in(H_x\cup H_z)^2} \left\|g_1^{(i)} - g_1^{(j)}\right\| \tag{3}$$

Following the same analysis, the updated models in the second iterations at our subject parameter servers are given by:

$$\theta_2^{(x)} = \theta_1^{(x)} - \gamma_2 MDA\left(g_2^{(1)}\ldots g_2^{(n_w)}\right)_x$$
$$\theta_2^{(z)} = \theta_1^{(z)} - \gamma_2 MDA\left(g_2^{(1)}\ldots g_2^{(n_w)}\right)_z$$

Thus, the difference between models now will be:

$$\left\|\theta_2^{(x)} - \theta_2^{(z)}\right\| = \left\|\left(\theta_1^{(x)} - \gamma_2 MDA\left(g_2^{(1)}\ldots g_2^{(n_w)}\right)_x\right) - \left(\theta_1^{(z)} - \gamma_2 MDA\left(g_2^{(1)}\ldots g_2^{(n_w)}\right)_z\right)\right\|$$
$$\leq \left\|\theta_1^{(x)} - \theta_1^{(z)}\right\| + \gamma_2 \left\|MDA\left(g_2^{(1)}\ldots g_2^{(n_w)}\right)_x - MDA\left(g_2^{(1)}\ldots g_2^{(n_w)}\right)_z\right\|$$

The bound on the first term is given in Equation 3 and that on the second term is given in Equation 2 and hence, the difference between models in the second iteration is given by:

$$\left\|\theta_2^{(x)} - \theta_2^{(z)}\right\| \leq 3 \cdot \gamma_1 \cdot \max_{(i,j)\in(H_x\cup H_z)^2} \left\|g_1^{(i)} - g_1^{(j)}\right\| + 3 \cdot \gamma_2 \cdot \max_{(i,j)\in(H_x\cup H_z)^2} \left\|g_2^{(i)} - g_2^{(j)}\right\| \tag{4}$$

By induction, we can write that the difference between models on two correct parameter servers at iteration $\tau$ is given by:

$$\left\|\theta_t^{(x)} - \theta_t^{(z)}\right\| \leq \sum_{t=1}^{\tau} 3 \cdot \gamma_t \cdot \max_{(i,j)\in(H_x\cup H_z)^2} \left\|g_t^{(i)} - g_t^{(j)}\right\| \tag{5}$$

Since $g_t^{(i)}$ and $g_t^{(j)}$ are computed at different workers, they can be computed based on different models $\theta_t^{(i)}$ and $\theta_t^{(j)}$. Following assumption 5, $\left\|g_t^{(i)} - g_t^{(j)}\right\|$ is bounded from above with $l\left\|\theta_t^{(i)} - \theta_t^{(j)}\right\|$. Noting that the sequence $\gamma_t$ is monotonically decreasing with $t \to \infty$ (Section 2.1), Equation 5 can be written as:

$$\left\|\theta_t^{(x)} - \theta_t^{(z)}\right\| \leq 3 \cdot \gamma_1 \cdot l \sum_{t=1}^{\tau} \max_{(i,j)\in(H_x\cup H_z)^2} \left\|\theta_t^{(i)} - \theta_t^{(j)}\right\|$$

Assuming that the maximum difference between any two correct models is bounded by $\mathcal{K}$, this difference can be written as:

$$\left\|\theta_t^{(x)} - \theta_t^{(z)}\right\| \leq 3 \cdot \gamma_1 \cdot l \cdot \mathcal{K} \cdot \tau$$

Now, to ensure the bound on the maximum difference between models, we need the value of $\left\|\theta_t^{(x)} - \theta_t^{(z)}\right\| \leq \mathcal{K}$. At this point, the number of steps $\tau = T$ should be bounded from above as follows:

$$T \leq \frac{1}{3 \cdot \gamma_1 \cdot l} \tag{6}$$

$T$ here represents the maximum number of iterations that are allowed to happen in the *scatter* step i.e., before doing one *gather* step. Doing more iterations than this number leads to breaking the requirement of *MDA* on the variance between input gradients, leading to breaking its Byzantine resilience guarantees. Thus, this bound is a *safety* bound that one should not pass to guarantee convergence. One can do less number of iterations (than $T$) during the *scatter* phase for a better performance (as we discuss in Section 6 in the main paper). Moreover, this bound requires that the initial setup satisfies the assumptions of *MDA*. Having a deployment that does not follow such assumptions leads to breaking guarantees of our protocol (as we show in Section 4 in the main paper).

## 3.4 Byzantine models filtering

This section shows that the filtering mechanism that is applied by workers (step 5 in Section 2.2) accepts only legitimate models that are received from servers which follow the algorithm, i.e., correct servers.

Such a filter is composed of two components: (1) a Lipschitz filter, which bounds the growth of models with respect to gradients, and (2) a models filter, which bounds the distance between models in two consecutive iterations. We first discuss the Lipschitz filter then the models filter; we show that using either of them only does not guarantee Byzantine resilience.

### 3.4.1 Lipschitz filter

The Lipschitz filter is defined in Step 5, Section 2.2. Roughly speaking, it runs on worker side, where it computes the Lipschitz coefficient of the pulled model and suspects it if its computed coefficient is far from Lipschitz coefficients of the previous correct models (from previous iterations).

The Lipschitz filter, by definition, accepts on average $n_{ps} - f_{ps}$ models each pulled $n_{ps}$ models. Such a bound makes sense given the round robin fashion of pulling models from servers (by workers) and the existence of (at most) $f_{ps}$ Byzantine servers. Based on this filter, each worker pulls, on average, $n_{ps} + f_{ps}$ each $n_{ps}$ iterations. Due to the presence of $f_{ps}$ Byzantine servers, this is a tight lower bound on the communication between each worker and parameter servers to pull the updated model. The worst attack an adversary can do is to send a model that passes the filter (looks like a legitimate model, i.e., very close to a legitimate model) that does not lead to computing a large enough gradient (i.e., leads to minimal learning progress); in other words: an attack that drastically slows down progress. For this reason, such a filter requires $n_{ps} > 3f_{ps}$. With this bound, the filter ensures the acceptance of at least $f_{ps} + 1$ models for each pulled $n_{ps}$ models, ensuring a majority of correct accepted models anyway and hence, ensuring the progress of learning. Moreover, due to the randomness of choosing the value $r_j$ and the round robin fashion of pulling the models, progress is guaranteed in such a step, as correct and useful models are pulled by other workers, leading to computing correct gradients.

Based on assumption 5 and the round robin fashion of pulling models, a Lipschitz coefficient that is computed based on a *correct* model is always bounded between two Lipschitz coefficients resulting from *correct* models. Based on this, the *global confinement* property is satisfied (based on the properties of models that passes the Lipschitz filter) and hence, we can plug LiuBei in the confinement proof in [3]. Precisely, given assumptions 4 and 5, and applying [3] (of which these assumptions are prerequisite), the models accepted by the Lipschitz filter satisfy the following property:

Let r=2,3,4, $\exists A'_r \geq 0$ and $B'_r \geq 0$ such that $\forall t \geq 0, \mathbb{E}\|g_t\| \leq A'_r + B'_r\|\theta_t\|^r$,

where $\theta_t$ is the model accepted by the Lipschitz filter at some worker and $g_t$ is the gradient calculated based on such a model.

### 3.4.2 Models filter

The Lipschitz filter is necessary, yet not sufficient to accept models from a parameter server. A Byzantine server can craft some model that is arbitrarily far from the correct model, whose gradient satisfies the Lipschitz filter condition. Thus, it is extremely important to make sure that the received model is close to the expected, correct model. Such a condition is verified by the second filter at the worker side, which we call the *models filter*. Such a filter is based on the results given in Section 3.3 and the local model estimation on the workers side.
Without loss of generality, consider a correct worker $j$ that pulls models from parameter servers $\theta_t^{(i)}\forall i \in [1\,..\,n_{ps}]$ in a round robin fashion. In each iteration $t > 1$, the worker computes a local estimation of the next (to be received) model $\theta_t^{(j(l))}$ based on the latest model it has $\theta_{t-1}^{(j)}$ and its own gradient estimation $g_{t-1}^{(j)}$. The local model estimation is done as follows:

$$\theta_t^{(j(l))} = \theta_{t-1}^{(j)} - \gamma_{t-1}g_{t-1}^{(j)}.$$

Without loss of generality, assume that worker $j$ pulls the model from some server $i$ in iteration $t$. If such a server is correct, it computes the new model $\theta_t^{(i)}$ as follows:

$$\theta_t^{(i)} = \theta_{t-1}^{(i)} - \gamma_{t-1}MDA\left(g_{t-1}^{(1)}\ldots g_{t-1}^{(n_w)}\right).$$

Thus, the difference between the local model estimation at worker $j$ and the received model from server $i$ (if it is correct) is given by:

$$\left\|\theta_t^{(j(l))} - \theta_t^{(i)}\right\| = \left\|\left(\theta_{t-1}^{(j)} - \gamma_{t-1}g_{t-1}^{(j)}\right) - \left(\theta_{t-1}^{(i)} - \gamma_{t-1}MDA\left(g_{t-1}^{(1)}\ldots g_{t-1}^{(n_w)}\right)\right)\right\|$$
$$\leq \left\|\theta_{t-1}^{(j)} - \theta_{t-1}^{(i)}\right\| + \gamma_{t-1}\left\|MDA\left(g_{t-1}^{(1)}\ldots g_{t-1}^{(n_w)}\right) - g_{t-1}^{(j)}\right\|$$

Based on the guarantees given by *MDA* [7], the following bound holds:

$$\left\|MDA\left(g_t^{(1)}\ldots g_t^{(n_w)}\right) - g_t^{(j)}\right\| \leq \frac{n_w - f_w}{2f_w}\left\|g_t^{(j)}\right\|.$$

Based on the results of Section 3.3, the maximum distance between two correct models just after a *gather* step is $\frac{3}{4}\frac{n_w - f_w}{f_w}\gamma_{(T\cdot(t \mod T))}\left\|g_{(T\cdot(t \mod T))}\right\|T$. By induction, we can find the bound on the value $\left\|\theta_{t-1}^{(j(l))} - \theta_{t-1}^{(i)}\right\|$ and hence, we can write:

$$\left\|\theta_t^{(j(l))} - \theta_t^{(i)}\right\| \leq \gamma_{(T\cdot(t \mod T))}\left\|g_{(T\cdot(t \mod T))}\right\|\left(2\left((t \mod T) - 1\right) + \frac{(n_w - f_w)(3T + 2)}{4f_w}\right). \tag{7}$$

Thus, a received model $\theta_t^{(i)}$ that satisfies Equation 7 is considered passing the models filter. Such a model is guaranteed to be in the correct cone of models in one *scatter* step. Note that such a filter cannot be used alone without the Lipschitz filter. A Byzantine server can craft a model that satisfies such a filter (i.e., the models filter) while being on the opposite direction of minimizing the loss function. Such a model then will be caught by the Lipschitz filter.
Combining the Lipschitz filter and the models filter guarantees that the received model at the worker side is close to the correct one (at the current specific *scatter* step), representing a reasonable growth, compared to the latest local model at the worker.

## 3.5 Proof of LiuBei

The algorithm works iteratively in two main phases, which we call *scatter* and *gather* phases. In the *scatter* phase, each parameter server works on its own local data without communicating with other servers. Also, workers pull the model from at most $f_{ps}+1$ servers in each iteration, without requiring to communicate with all other servers. Such a step lasts for $T$ learning iterations. In the *gather* phase, parameter servers communicate together to gather their models close to each other, and workers also gather models from all parameter servers. Such a step is done for only one learning iteration. Naively, if we do the *gather* step in all learning iterations, the algorithm becomes very similar to *GuanYu* [6] (with even better guarantees due to the network synchrony assumption), and the proof of convergence can be adapted directly (basically in this case we do not need the filters). On the other extreme, if the *gather* step is never executed (i.e., only the *scatter* step is executed), the models at different servers will become arbitrarily far form each other (Section 3.3) and hence, the assumptions of the Byzantine-resilient GAR, i.e., *MDA* are violated, leading to (possibly) divergence.

Based on this, the gist of the proof is to show that the maximum distance between models at correct servers is always small enough to satisfy the assumptions of *MDA*. Based on this and the fact that workers do SGD steps during the *scatter* phase, it is trivial to show that there is a progress in learning between two *gather* steps, i.e., the loss function is minimized and hence, the proof is reduced to the standard proof of SGD convergence [3].

Formally, we show:

$$\lim_{t \to \infty} \mathbb{E} \left( \max_{(x,z) \in [1 \,..\, n_{ps} - f_{ps}]} \left\| \theta_t^{(x)} - \theta_t^{(z)} \right\| \right) = 0$$
$$\lim_{t \to \infty} \mathbb{E} \left\| \nabla L \left( \theta_t^{(x)} \right) \right\| = 0$$

Proving these two equalities implies necessarily the convergence of the learning procedure, or formally:

$$\lim_{t \to \infty} \mathbb{E} \left\| \nabla L \left( \theta_t \right) \right\| = 0$$

*Proof.* From Equation 7, we know that the difference between correct models is bounded by $\left\| g_{T \cdot (t \mod T)} \right\|$. Based on our assumptions (Section 3.1), $\lim_{t \to \infty} \left\| g_{T \cdot (t \mod T)} \right\| = 0$.

Therefore, $\lim_{t \to \infty} \max_{(x,z) \in [1 \,..\, n_{ps} - f_{ps}]} \left\| \theta_t^{(x)} - \theta_t^{(z)} \right\| = 0$. $\qquad \square$

A direct consequence of Section 3.3 is that beyond a certain time $t_\epsilon$ (with $\varepsilon > 0$ be any positive real number), the standard deviation of the gradient estimators as well as the *drift* between (correct) parameter vectors can be bounded arbitrarily close to each other. More formally, the following holds (and is a direct consequence of the limits stated above):

$$\exists t_\varepsilon \geq 0, \ \forall t \geq t_\varepsilon, \ \begin{cases} \mathbb{E} \left( \max_{i \in [1 \,..\, n_w - f_w]} \left\| g_t^{(i)} - \mathbb{E} \, g_t^{(1)} \right\| \right) \leq \sigma' + \varepsilon \\[2mm] \mathbb{E} \left( \max_{(x,z) \in [1 \,..\, n_{ps} - f_{ps}]^2} \left\| \theta_t^{(x)} - \theta_t^{(z)} \right\| \right) < \frac{\varepsilon}{l} \end{cases}$$

The first inequality provides the bounded variance guarantee needed to plug LiuBei into the convergence proof of [2], and the second inequality provides the remaining requirement, i.e. the bound on the statistical higher moments of the gradient estimator as if there was a *single* parameter. More precisely, we have the following, let $(x, z)$ be any two correct parameter servers, using a triangle inequality, the second inequality above, and assumption 4 (bounded moments) we have (as given in Section 3.4):

$$\forall t > t_\varepsilon \forall r \in [2 \,..\, 4], \ \exists (A_r, B_r) \in \mathbb{R}^2,$$
$$\forall (i, t, \theta) \in [1 \,..\, n_w - f_w] \times \mathbb{N} \times \mathbb{R}^d, \ \mathbb{E} \left\| g_t^{(x)} \right\| \leq A'_r + B'_r \left\| \theta_t^{(z)} \right\|^r$$

where $A_r'$ and $B_r'$ are positive constants, depending polynomially (at most with degree $r$, by using the second inequality above, the triangle inequality and a binomial expansion) on $\varepsilon$, $\sigma$ and $(A_r, B_r)$ from assumption 4, allowing us to use the convergence proof of [2] (Proposition 2) regardless of the identity of the (correct) parameter server.

**Convergence rate.** Using *MDA* is shown to induce a convergence rate of $\mathcal{O}(\sqrt{\frac{n_w - f_w}{n_w}})$ [7]. We note that if we do the *gather* step in each iteration (i.e., $T = 1$), this rate would not change since the *MeaMed* function will always choose a correct model to proceed with. Moreover, the variance between models at the correct parameter servers will be minimum (zero in the case of absence of Byzantine machines). Gathering the models each 2 iterations (i.e., $T = 2$) doubles the distance between models on correct parameter servers (in the worst case). By induction, doing the *gather* step after $T$ iterations stretches the difference between correct models by a factor of $T$ in the worst case. Hence, the slowdown induced by the filtering techniques only is $\mathcal{O}(\frac{1}{T})$ and the convergence rate of LiuBei is $\mathcal{O}(\sqrt{\frac{n_w - f_w}{n_w T^2}})$.

# 4 Validating the bounded gradients variance assumption

To make progress at every step, any state–of–the–art Byzantine–resilient gradient aggregation rule (GAR), that is based *solely* on statistical robustness, requires a bound on the ratio *variance*/*norm* of the correct gradient estimations. Intuitively, not having such a bound would allow the correct gradients to become *indistinguishable* from some random noise. This is problematic, since these Byzantine–resilient GARs [2, 10, 7, 12] rely on techniques analogous to *voting* (i.e. median–like techniques in high–dimension): if the correct majority does not agree (appears "random"), then the Byzantine minority controls the aggregated gradient. For example, not satisfying these bounds makes the used GARs vulnerable against recently–proposed attacks like *A little is enough* attack [1], which we experimented in Section 4.3 in the main paper. Such a bound is to ensure that, no matter the received Byzantine gradients, the expected value of the aggregated gradient does lie in the same half–space as the real gradient, leading for every step taken to more *optimal* parameters (smaller loss).

Here, we try to understand when this assumed bound on the variance to norm ratio (e.g., Equation 1) holds, and when it does not. The most straightforward way to fulfill such an assumption is to increase the batch size used for training. The question is then what the minimum batch size (that can be used while satisfying such a bound) is, and whether it is small enough for the distribution of the training to still make sense.

**Methodology.** We use the same setup, along with hyper-parameters, used in our evaluation of LiuBei (Section 4.2). We estimate over the first 100 steps of training the variance to norm ratio of correct gradient estimations for several batch sizes. We plot the average (line) and standard deviation (error bar) of these ratios over these 100 steps (Figure 1). We show the bound required by two Byzantine-resilient GAR: *MDA*, and *Multi–Krum* [2]. We find *Multi–Krum* a very good example on a widely–used GAR that unfortunately does not seem to provide any practical[3] guarantee, due to its unsatisfied assumption[4]. We also experimented with two values of the number of declared Byzantine workers: $f = 1, 5$. Increasing the value of $f$ calls for a tighter bound on the variance to norm ratio.

**Results.** Figure 1 depicts the relation between the variance (of gradients) to norm ratio with the batch size. According to such a figure, *Multi–Krum* cannot be safely used even with the largest experimented batch size, i.e., 256. Otherwise, the variance bound assumption such a GAR builds on is not satisfied and hence, an adversary can break its resilience guarantees [1]. *MDA* gives a better bound on the variance, which makes it more practical in this sense: typical batch size of 128 for example can be safely used with $f = 1$. However, *MDA* is not safe to

---

[3] At least on our academic model and dataset.

[4] It needs very low variance to norm ratio of correct gradient estimations, e.g. 0.08 for $(n, f) = (18, 5)$.

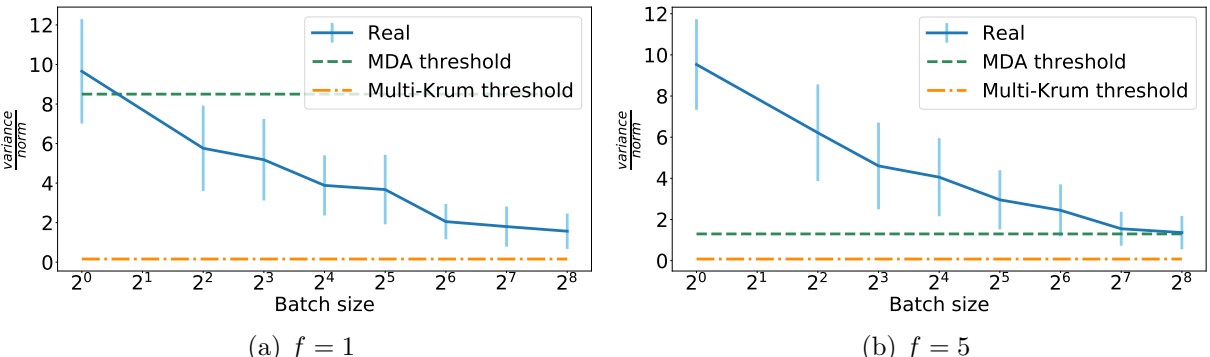

(a) $f = 1$                     (b) $f = 5$

Figure 1: Variance to norm ratio with different batch sizes, compared to the bound/threshold required by two Byzantine-resilient GARs: *MDA* and *Multi–Krum*. To satisfy a GAR assumption/condition, the *real* $variance/norm$ value should be lower than the GAR bound. For instance, *MDA* can be used with batch-size=32 with $f = 1$, but not with $f = 5$ (as the *real* $variance/norm$ value is higher than the *MDA* threshold).

use with $f = 5$ even with the largest experimented batch size ($b = 256$). This is confirmed in Section 4.3 in the main paper, where we show that an adversary can use such a vulnerability (due to the unsatisfied assumption) to reduce the learning accuracy. Having the optimal bound on variance while guaranteeing Byzantine resilience and convergence remains an open question.

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
