# OpenReview forum: "Fast Machine Learning with Byzantine Workers and Servers"
_ICLR.cc/2020/Conference — Reject_

### Official Review · AnonReviewer2 · 2019-10-23
**Official Blind Review #2**

**Rating:** 6

**Review:**

This paper introduces an algorithm to build distributed SDG-based training algorithm that are robust to Byzantine workers and servers.

I am not very familiar with this area of research, but I feel the authors did a good job providing clear explanations and introducing all the relevant concepts needed to understand the proposed algorithm. Overall, I found the paper an interesting read.

The experimental section of the paper is lacking in some aspects:
- One of the main ideas introduced in the paper is that of filters to check the legitimacy of models from model servers. While these ideas are sensible from a technical point of view, I feel the experimental section is not properly demonstrating all the robustness claims made in the paper. For example, in the beginning of training with high learning rates the models will change a lot, are these filters effective in this situation as well? How are these filters working in terms of false positive/negatives in the experiments?
- How are models corrupted during training? What's the performances of the filters with different corruption techniques (e.g. adversarial attacks)?
- What's the impact of the choice of T in the experiments?

**Experience Assessment:**

I do not know much about this area.

**Review Assessment: Checking Correctness Of Derivations And Theory:**

I assessed the sensibility of the derivations and theory.

**Review Assessment: Checking Correctness Of Experiments:**

I assessed the sensibility of the experiments.

**Review Assessment: Thoroughness In Paper Reading:**

I read the paper at least twice and used my best judgement in assessing the paper.

---

> ### Author Response · Authors · 2019-11-08
> **We thank AnonReviewer2 for the comments. We answer below to the points raised by the reviewer**
>
> “One of the main ideas introduced in the paper is that of filters to check the legitimacy of models from model servers. While these ideas are sensible from a technical point of view, I feel the experimental section is not properly demonstrating all the robustness claims made in the paper.”
> >> To experiment robustness against Byzantine workers, we did experiments with a state-of-the-art attack coined as “A little is enough attack” [1]. If the reviewer is talking about experiments with Byzantine servers, then we will add a few Byzantine behaviors on the servers’ side, and we will report the algorithm convergence in the updated version of the paper, which we plan to have before the end of the rebuttal period.
> It is important to note however that in the case of Byzantine failures, which by definition can be arbitrary, the standard guarantee is to provide formal proof (which we did), experiments are only for illustrative purposes.
>
> “For example, in the beginning of training with high learning rates the models will change a lot, are these filters effective in this situation as well? How are these filters working in terms of false positive/negatives in the experiments?”
> >> We designed our filters to be adaptive to the learning status. Concretely, the filters adapt their behavior based on the learning rate and the gradient norm of the current scatter step. Based on that, at the beginning of learning where the learning rate is high, such filters accept a wider range of models. Later on, this window decreases with the learning convergence. Our analysis guarantees the absence of false positives, i.e., no corrupted model will pass the filter; this is the safety guarantee that our algorithm provides. Nonetheless, showing empirically false positives/negatives of our filters is an interesting aspect that we are planning to add to the paper before the deadline.
>
> “How are models corrupted during training? What's the performances of the filters with different corruption techniques (e.g. adversarial attacks)?”
> >> In theory, the models could be corrupted in any means, ranging from dropping some values in the model weights to carefully crafting corrupted models to diverge the learning procedure. We will report the behavior of our algorithm under different models’ corruption (i.e., Byzantine servers) in the updated version of the paper.
>
> “What's the impact of the choice of T in the experiments?”
> >> In theory, the higher the T, the better the throughput, the less the learning quality. The tradeoff between the quality of models on servers (how far they are from each other) and the throughput is very interesting to study. We will add experiments to show this in the updated version of the paper.
>
> [1] Baruch, Moran, Gilad Baruch, and Yoav Goldberg. "A Little Is Enough: Circumventing Defenses For Distributed Learning." arXiv preprint arXiv:1902.06156 (2019).

---

> > ### Comment · AnonReviewer2 · 2019-11-13
> > **Thanks for the new results**
> >
> > I have seen the revised version of the paper, the new results look very interesting. Overall, I think Weak Accept is a fair score for this paper, so I will leave it unchanged and argue for acceptance.

---

> > > ### Author Response · Authors · 2019-11-14
> > > **Thank you**
> > >
> > > Thank you very much for your time. We will be happy to address any other concern you might raise after the discussion with the other reviewers.

---

### Official Review · AnonReviewer1 · 2019-10-28
**Official Blind Review #1**

**Rating:** 3

**Review:**

The paper considers distributed stochastic gradient descent, where some (unknown) compute nodes may be unreliable. New heuristics for filtering out replies from unreliable servers are introduced alongside a new protocol that helps keeping nodes in sync.

In general, I miss a more clear indication of how the individual contributions are different from other methods. I am also missing more detailed ablation studies showing which of the new ideas contribute the most to efficient learning. As far as I can tell, the experiments do not really show an improvement over existing methods in this domain.

This is not my area of expertise, but I cannot recommend the paper for publication in its current form as
(a) it's not clear to me that the paper improves on existing methods, and
(b) it's not clear to me what the real novelty of the work is.

Post-rebuttal:
I acknowledge the response of the authors. They clarified some aspects for me, and the paper appears to have improved over the rebuttal period.
I did not change my rating, but I want to emphasize that this is only because my knowledge of this field is so limited. My rating is largely based on "gut feeling" rather than actual knowledge, and I won't argue against acceptance.

**Experience Assessment:**

I do not know much about this area.

**Review Assessment: Checking Correctness Of Derivations And Theory:**

N/A

**Review Assessment: Checking Correctness Of Experiments:**

I assessed the sensibility of the experiments.

**Review Assessment: Thoroughness In Paper Reading:**

I made a quick assessment of this paper.

---

> ### Author Response · Authors · 2019-11-08
> **We thank AnonReviewer1 for the comments. We answer below to the points raised by the reviewer**
>
> “In general, I miss a more clear indication of how the individual contributions are different from other methods. it's not clear to me what the real novelty of the work is.”
> >> We would like to clarify the main contributions of this work. First, utilizing filtering techniques to tolerate Byzantine servers is novel as previous proposals use robust aggregation to do so. This goes into designing a novel filtering mechanism, which we call models filter, in addition to the novel adaptation of the Lipschitz filter idea to tolerate Byzantine models. For this, we design the local, speculative step that should be done by workers in order to compute the Lipschitz coefficient correctly; this last step is novel in this work. Second, we propose a communication protocol, which we call scatter/gather, to control the communication between servers and workers. Such a protocol is also novel, and it (along with the proposed filtering components) contributes to reducing the communication overhead by not only reducing the number of required communication rounds per iteration but also with reducing the number of communicated messages per round. Third, we theoretically prove that our algorithm, along with the proposed communication protocol, guarantees Byzantine behavior tolerance and learning convergence. Finally, we empirically show the performance of our algorithm compared to two baselines in practical setups, where we discuss convergence overhead, Byzantine tolerance, and system’s throughput.
>
> “I am also missing more detailed ablation studies showing which of the new ideas contribute the most to efficient learning.”
> >> We achieve efficient learning mainly by reducing the communication overhead by reducing both (1) the number of communication rounds per iteration and (2) the number of messages per communication round. This is enabled by the novel idea of using filtering instead of robust aggregation to tolerate Byzantine servers in addition to the design of our scatter/gather protocol. Both help drastically reduce the communication overhead and contribute to the learning efficiency.
> An interesting experiment that we are working on now (and planning to add it to the paper before the deadline) is to show the effect of the value of T (the number of learning iterations per one scatter step) on the learning convergence. We believe such an experiment will also shed light on the inherent tradeoff between the learning quality and the system’s throughput.
>
> “the experiments do not really show an improvement over existing methods in this domain. it's not clear to me that the paper improves on existing methods.”
> >> To show the improvement of LiuBei over existing methods, we evaluate it against two baselines: TensorFlow and GuanYu [2]. Compared to TensorFlow, LiuBei guarantees tolerance to both Byzantine servers and workers. Compared to GuanYu, LiuBei offers a 70% throughput gain with the same convergence behavior. Compared to other Byzantine tolerant algorithms, LiuBei offers an additional tolerance to Byzantine servers, which were assumed trusted and correct in such algorithms.
>
> [1] Georgios Damaskinos, et al. "Asynchronous Byzantine machine learning (the case of SGD)." ICML'18.
> [2] El-Mhamdi, El-Mahdi, et al. "SGD: Decentralized Byzantine Resilience." arXiv preprint arXiv:1905.03853 (2019).
>
> Post rebuttal:
> Thank you very much for your time. We will be happy to address any other concerns you might raise after the discussion with the other reviewers.

---

### Official Review · AnonReviewer3 · 2019-10-29
**Official Blind Review #3**

**Rating:** 3

**Review:**

My review got deleted because the title kept creating an unexplained error.
Here's another shorter attempt

I haven't really followed along the literature for this. But from the results, it's not immediately clear to me what practical setup this is useful in. The authors assume perfect network synchrony, they have a 25% overhead on TensorFlow and they have a comparison to another algorithm that operates under different assumptions.

Who would ever use this and why? What's the plan for getting data to the untrusted workers?

**Experience Assessment:**

I do not know much about this area.

**Review Assessment: Checking Correctness Of Derivations And Theory:**

I assessed the sensibility of the derivations and theory.

**Review Assessment: Checking Correctness Of Experiments:**

I assessed the sensibility of the experiments.

**Review Assessment: Thoroughness In Paper Reading:**

I made a quick assessment of this paper.

---

> ### Author Response · Authors · 2019-11-08
> **We thank AnonReviewer3 for the comments. We answer below to the points raised by the reviewer**
>
> “It's not immediately clear to me what practical setup this is useful in. The authors assume perfect network synchrony. Who would ever use this and why? What's the plan for getting data to the untrusted workers?”
> >> As a practical setup, think of a hospital for example that runs an ML application to help doctors give medications to their patients [7]. To accommodate for the huge data such a hospital gathers from patients and for the complex models it trains to achieve high accuracy for such a sensitive task, the hospital distributes learning on multiple machines [1]. For increased security, these machines run different implementations of the code (for the model training) [8]. The synchrony assumption could be achieved in such a controlled environment, i.e., engineers can expect an upper bound on the communication and computation delays. Several kinds of failures could happen to this setup, ranging from software bugs to adversarial behavior resulting from hacks to these machines. Moreover, the gathered data could be sometimes misleading or incomplete, which may pose a critical threat to training the hospital ML model. Another avenue where tolerating data from untrusted workers is of growing interest is on-device ML [2].
> We believe that the distributed ML literature, and specifically the Byzantine ML literature (e.g., [3-6] to name a few), also focused on environments with network synchrony. We take these efforts one step forward and address the inevitable case of server’s failures. We provide a proven solution to this problem which guarantees not only tolerance to such failures but also the convergence of the training procedure.
>
> “ they have a 25% overhead on TensorFlow”
> >> We believe that 24% of convergence overhead, compared to vanilla TensorFlow, is acceptable in the Byzantine ML literature. For instance, AggregaThor [9], a state-of-the-art system that tolerates only Byzantine workers, reports 19% to 43% convergence overhead. GuanYu [6], the only existing algorithm to tolerate Byzantine workers and servers, reports a 30% convergence overhead. Based on that, we believe the performance of LiuBei lies in the typical range of the overhead achieved by similar algorithms, also keeping in mind the strong guarantees LiuBei provides.
>
> “they have a comparison to another algorithm that operates under different assumptions.”
> >> We confirm that GuanYu, the main baseline, was designed to be used in a different environment than what we are considering in this paper (we assume network synchrony while GuanYu assumed network asynchrony). Yet, we compare with GuanYu as it is the only proposal, to the best of our knowledge, that addresses Byzantine resilience to both servers and workers. We would also like to clarify that we run GuanYu in a synchronous environment in our experiments to maintain comparison fairness. We also compare with vanilla TensorFlow that assumes network synchrony, yet does not tolerate Byzantine workers nor servers.
>
> [1] Li, Mu, et al. "Scaling distributed machine learning with the parameter server." OSDI'14.
> [2] Konečný, Jakub, et al. "Federated learning: Strategies for improving communication efficiency." arXiv preprint arXiv:1610.05492 (2016).
> [3] Chen, Lingjiao, et al. "Draco: Byzantine-resilient distributed training via redundant gradients." arXiv preprint arXiv:1803.09877 (2018).
> [4] Xie, Cong, et al. "Zeno: Byzantine-suspicious stochastic gradient descent." arXiv preprint arXiv:1805.10032 (2018).
> [5] Zhixiong Yang et al. "BRIDGE: Byzantine-resilient Decentralized Gradient Descent." arXiv preprint arXiv:1908.08098 (2019).
> [6] El-Mhamdi, El-Mahdi, et al. "SGD: Decentralized Byzantine Resilience." arXiv preprint arXiv:1905.03853 (2019).
> [7] Esteva, Andre, et al. "Dermatologist-level classification of skin cancer with deep neural networks." Nature 542.7639 (2017): 115.
> [8] Castro, Miguel, and Barbara Liskov. "Practical Byzantine fault tolerance." OSDI' 99.
> [9] Georgios Damaskinos, et al. "AGGREGATHOR: Byzantine Machine Learning via Robust Gradient Aggregation." SysML'19.

---

### Author Response · Authors · 2019-11-13
**Uploaded a revised version based on the reviewers' recommendations**

Based on the reviewers' recommendations, we uploaded a new version of our paper that describes a few additional experiments. We added the following experiments to the paper:

1) We show the performance of LiuBei, our algorithm, with Byzantine servers. We experimented with 4 Byzantine behavior and showed that LiuBei converges safely despite such a behavior.
2) We show the effect of different values for T on the convergence. The main finding is: the smaller the value of T, the slower the algorithm is yet, the more robust it is.
3) We show the percentage of false positives and false negatives produced by the filters of LiuBei. We show that our filters do not have false positives (i.e., the filters never pass a Byzantine model) while keeping the ratio of false negatives < 1% with different values for T.

---

### Decision · Program_Chairs · 2019-12-19

**Decision:**

Reject

**Comment:**

This paper is concerned with learning in the context of so-called Byzantine failures. This is relevant for for example distributed computation of gradients of mini-batches and parameter updates. The paper introduces the concept and Byzantine servers and gives theoretical and practical results for algorithm for this setting.

The reviewers had a hard time evaluating this paper and the AC was unable to find an expert reviewer. Still, the feedback from the reviewers painted a clear picture that the paper did not do enough to communicate the novel concepts used in the paper.

Rejection is recommended with a strong encouragement to use the feedback to improve the paper for the next conference.